# Who is engaging with lateral flow testing for COVID-19 in the UK? The COVID-19 Rapid Survey of Adherence to Interventions and Responses (CORSAIR) study

Louise E Smith [1,2] Henry WW Potts [3] Richard Amlôt [2,4] Nicola T Fear [1,5] Susan Michie [6] G James Rubin [1,2]

For numbered affiliations see end of article.

**Correspondence to**
Dr Louise E Smith;
louise.e.smith@kcl.ac.uk

## ABSTRACT

**Objectives** To investigate uptake of lateral flow testing, reporting of test results and psychological, contextual and socio-demographic factors associated with testing.
**Design** A series of four fortnightly online cross-sectional surveys.
**Setting** Data collected from 19 April 2021 to 2 June 2021.
**Participants** People living in England and Scotland, aged 18 years or over, excluding those who reported their most recent test was a polymerase chain reaction (PCR) test (n=6646, n≈1600 per survey).
**Main outcome measures** Having completed at least one lateral flow test (LFT) in the last 7 days.
**Results** We used binary logistic regressions to investigate factors associated with having taken at least one LFT. Increased uptake of testing was associated with being vaccinated (adjusted ORs (aORs)=1.52–2.45, 95% CI 1.25 to 3.07, analysed separately by vaccine dose), employed (aOR=1.94, 95% CI 1.63 to 2.32), having been out to work in the last week (aOR=2.30, 95% CI 1.94 to 2.73) and working in a sector that adopted LFT early (aOR=2.54, 95% CI 2.14 to 3.02) . Uptake was higher in people who reported cardinal COVID-19 symptoms in the last week (aOR=1.89, 95% CI 1.34 to 2.66). People who had heard more about LFTs (aOR=2.28, 95% CI 2.06 to 2.51) and knew they were eligible to receive regular LFTs (aOR=2.98, 95% CI 2.35 to 3.78) were also more likely to have tested. Factors associated with not taking a test included agreeing that you do not need to test for COVID-19 unless you have come into contact with a case (aOR=0.51, 95% CI 0.47 to 0.55).
**Conclusions** Uptake of lateral flow testing is low. Encouraging testing through workplaces and places of study is likely to increase uptake, although care should be taken not to pressurise employees and students. Increasing knowledge that everyone is eligible for regular asymptomatic testing and addressing common misconceptions may drive uptake.

## INTRODUCTION

As the UK moves to continuous management of COVID-19 instead of disaster prevention, a variety of strategies are being used to slow

## STRENGTHS AND LIMITATIONS OF THIS STUDY

⇒ A series of large cross-sectional surveys investigating uptake, and factors associated with uptake, of lateral flow testing in the English and Scottish population.
⇒ One of few studies investigating psychological, contextual and socio-demographic factors associated with uptake of testing.
⇒ Data were collected in the weeks immediately after the recommendation that all UK adults should take two rapid lateral flow tests (LFTs) per week.
⇒ Self-reported rates of having completed an LFT in the last week were higher than estimated by official agencies.
⇒ The behaviour and beliefs of people who have signed up to complete internet surveys may not be representative of those of the general population.

the spread of infection. It is thought that transmission while asymptomatic accounts for 24% of COVID-19 transmission.[1] There is limited evidence of the effectiveness of mass asymptomatic testing programmes at slowing the spread of COVID-19 as most programmes, both in the UK and abroad, have been used in conjunction with other behavioural restrictions, making it impossible to quantify the impact of mass testing alone.[2 3] However, the effectiveness of any intervention will be limited if people do not engage with that behaviour.

Since 9 April 2021, everyone in the UK has been able to access free, regular, rapid lateral flow COVID-19 testing for use when asymptomatic.[4] At the time of writing, the English and Scottish Governments are recommending twice weekly lateral flow testing for all adults. Lateral flow tests (LFTs) for asymptomatic testing can be ordered online, can be collected from NHS pharmacies and are

supplied through schools, colleges and nurseries; some universities and other employers also offer rapid testing.[5] Results of these tests should be reported through a UK Government website.[6] Mass asymptomatic testing was piloted in Liverpool, England, between 6 November 2020 and 30 April 2021. Findings indicated that 57% of residents in Liverpool took at least one rapid LFT over the course of the pilot (~6 months), but uptake was substantially lower in more deprived areas (where infection rates were higher) and among non-white minoritised ethnic groups.[7 8] Testing in the initial months was partly driven by the intense media campaign in Liverpool, and the novelty of testing at the time. As well as asymptomatic testing being available, all those with cardinal COVID-19 symptoms (high temperature, a new, continuous cough and a loss or change to sense of smell or taste) have been eligible for a polymerase chain reaction (PCR) test since 18 May 2020.[9] There is considerable confusion among the public about the use of LFTs and PCR tests when symptomatic.[10] Previous research indicates that uptake of testing when symptomatic is low.[11]

A range of factors may affect whether people engage with lateral flow testing. These can be categorised using the Capability, Opportunity, Motivation and Behaviour (COM-B) model.[12] Capability encompasses the psychological and physical capacity to engage in a behaviour. It includes, for example, knowledge as to what the appropriate behaviour is (eg, knowing that you are eligible for regular testing) and when to enact it. Opportunity relates to factors outside the person, for example, testing being required by one's employer, or belonging to a group that was eligible for asymptomatic testing prior to the nationwide rollout (eg, health and social care workers, teachers, students and people who work in transport such as hauliers)[13–16] and that, in turn, may be associated with socioeconomic status or ethnicity. Motivation describes the psychological processes that energise or inhibit a behaviour and includes perceived risk associated with a disease outbreak, which may in turn be linked to greater exposure to other people (eg, during socialising),[17] believing that you have immunity against COVID-19,[18] believing that you could engage in a behaviour if you wanted to (self-efficacy) and perceiving the behaviour to be effective.

The aim of this study was to investigate rates of uptake of lateral flow testing and reporting of results in England and Scotland, and the psychological, contextual and socio-demographic factors associated with testing.

## METHODS
### Design
From January 2020, BMG Research was conducting a series of nationally representative (UK) cross-sectional surveys (weekly or fortnightly) on behalf of the Department of Health and Social Care throughout the COVID-19 outbreak. We analysed these data as part of the COVID-19 Rapid Survey of Adherence to Interventions and Responses (CORSAIR) study.[11] For this study, we used data collected between 19 April 2021 and 2 June 2021 (rounds 48–51).

### Participants
Participants (n≈2000 per survey) were eligible for the study if they were aged 16 years or over and lived in the UK, and were recruited from two specialist research panel providers, Respondi (n=50 000) and Savanta (n=31 500). Quotas were applied based on age and gender (combined) to try to match these characteristics to those of the general UK population structure. Consent was implied by participants' completion of the survey, in line with industry standards. After completing the survey, participants were then unable to participate in the subsequent three surveys; thus, all participants included in this study were unique. Participants were reimbursed in points which could be redeemed in cash, gift vouchers or charitable donations (points for this survey had a monetary value of up to 70p).

We limited the sample to people living in England or Scotland as Wales and Northern Ireland were following a different testing schedule. We excluded those under 18 years of age as many would be eligible for asymptomatic testing under school testing regimes.

To investigate uptake of lateral flow testing, we excluded people who reported that their most recent test was a PCR test or they did not know what type their most recent test was and who reported that they had completed a PCR test or who did not know what type of test they completed after developing COVID-19 symptoms.

### Study materials
#### Lateral flow testing
Participants were asked, 'when was the last time [they] had a test for coronavirus'. Response options ranged from 'within the last 24 hours' to 'I've never had a coronavirus test'.

People who reported having a COVID-19 test in the last week were asked a series of follow-up questions. These included: how many times they had taken a COVID-19 test in the last 7 days (responses ranged from 'once' to '10 times or more'); how they received their most recent COVID-19 test (response options included receiving it from a care home, one's place of work, a school, further education college or university, a hospital/clinical setting, having ordered it online, collecting a pack from a local test site or taking an assisted test at a local test site, or when travelling internationally); where they reported their test result, if at all (response options included the GOV.UK website, by phone with NHS Test and Trace, to one's employer, a school, further education college or university, and to friends and family); what the result of their most recent test was (response options included 'I tested positive', 'I tested negative', 'The test was void (inconclusive)' and 'I have not received my results yet'); and which type of test they had most recently taken (response options included 'PCR test', 'lateral flow test' or 'don't

know/unsure'). For this question, participants were given an explanation of each test ('PCR tests are sent to a lab for processing and results are sent to you usually by text or email. This includes home test kits which you need to mail in or drop off. Rapid lateral flow tests provide results within 30 min of taking the test (these might also be referred to as rapid antigen tests). Both tests involve swabbing the back of your nose and throat').

All participants were asked how much they had previously heard about 'free, regular rapid testing for people even if they don't have coronavirus symptoms, which uses a technology called 'lateral flow testing'' on a 4-point scale from 'nothing at all' to 'a great deal'. They were also asked if 'as far as [they knew, they were] eligible to receive rapid COVID-19 tests twice a week to check for coronavirus even if [they didn't] have symptoms (also known as lateral flow testing)?'. Possible responses were 'yes', 'no' and 'don't know'.

## Contextual factors

All participants were asked if they had any symptoms in the past 7 days from a list of 10 (new, continuous cough; high temperature/fever; runny nose; diarrhoea; nausea/feeling sick; vomiting; sneezing; loss of appetite; loss of sense of smell; and loss of taste). In the final round of data collection, this question was changed to ask if participants had developed any symptoms in the past 10 days (response options remained identical).

We identified people in groups who had been eligible for asymptomatic testing before it became available to everyone (people who indicated that they worked or volunteered in health or social care, education and childcare, or transport). Participants were identified as students if they specified that their employment status was 'student/on a government training programme (Nation Traineeship/Modern Apprentice)'.

Participants were asked how many times in the last 7 days they had been out of their home to go to work and to meet up with friends or family that they did not live with (responses capped at 30). In the last round of data collection, this question was changed to ask how many times in the past 7 days they had 'left the house to go out to work (number of days)' and 'met up with' friends or family they did not live with (responses capped at 30). We recoded the number of times people had been out for work into a binary variable (been out to work in the last week vs not).

## Psychological factors

Participants were asked how much, if at all, they agreed that: they were confident that LFTs were accurate; regularly testing people without symptoms was an effective way to prevent COVID-19 transmission; they did not need to take an LFT unless they had come into contact with a COVID-19 case; and people who had been vaccinated did not need to be tested for COVID-19 regularly. All questions were asked on a scale from 'strongly disagree' to 'strongly agree'.

Perceived risk of COVID-19 was measured by asking participants to what extent they thought COVID-19 posed a risk to 'you personally' and 'people in the UK' on a 5-point scale from 'no risk at all' to 'major risk'.

## Socio-demographic factors

Participants were asked for their sex, age, employment status, highest educational or professional qualification, ethnicity, first language and COVID-19 vaccination status. Participants were also asked whether they had any dependent children in their household, whether they or a member of their household had a chronic illness, and whether the highest earner in their household worked in a manual occupation. We computed a quadratic term for age. Region and index of multiple deprivation were derived from participants' postcode.[19]

Participants were also asked if they had had, or currently had, COVID-19. We recoded responses to give a binary variable (think have had COVID-19 ('I've definitely had it, and had it confirmed by a test' and 'I think I've probably had it') vs think have not had COVID-19 ('I don't know whether I've had it or not', 'I think I've probably not had it' and 'I've definitely not had it')).

We measured financial hardship by asking participants to what extent in the past 7 days they had been struggling to make ends meet, skipping meals they would usually have and were finding their current living situation difficult (Cronbach's $\alpha$=0.83).

## Patient and public involvement

Lay members served on the advisory group for the project that developed our prototype survey material; this included three rounds of qualitative testing.[20] Due to the rapid nature of this research, the public was not involved in the further development of the materials during the COVID-19 pandemic.

## Power

We used a post-hoc power calculation to calculate power achieved in logistic regression analyses. With a sample size of 6646, we had 100% power to detect small ORs (OR=1.67)[21] at $\alpha$=0.003 (probability of taking an LFT in last week=0.169).

## Analysis

We report uptake of lateral flow testing, how people received their most recent test and out-of-home behaviour following a positive test result descriptively.

Contextual, psychological and socio-demographic factors were identified a priori, based on theory (COM-B model).[12] We ran multivariable logistic regression analyses to investigate contextual, psychological and socio-demographic factors associated with having completed at least one COVID-19 test in the last week, controlling for survey round, region, gender, age (raw and quadratic), the presence of a dependent child in the household, having a chronic illness oneself, having a household member who has chronic illness, employment status, highest earner in the household working in a manual occupation,

index of multiple deprivation quartile (2019),[19] highest educational or professional qualification, ethnicity, first language, having had COVID-19 before, vaccination status and financial hardship. For these analyses, we excluded people who reported that they did not know when their last COVID-19 test was. As we hypothesised that peoples' beliefs about the necessity for regular asymptomatic testing of people who had been vaccinated for COVID-19 might differ based on vaccination status, we investigated associations between this belief and increased uptake of testing separately in those who reported having no, one or two doses of a COVID-19 vaccine. We also hypothesised that participants who received a positive COVID-19 test or whose test result was inconclusive may perceive the risk of COVID-19 to themselves differently to those who had received a negative test. Therefore, we also investigated associations between perceived risk to oneself and increased uptake of testing, excluding people whose most recent test result was positive or inconclusive.

Due to the large number of analyses (n=16) conducted on a single outcome, we used a Bonferroni correction and only report narratively results where p<0.003. Tables give raw p values.

We conducted a sensitivity analysis, running logistic regression analyses using index of multiple deprivation as a categorical variable.

## RESULTS

### Participant characteristics

Of participants included in analyses, 54% were female, with participants having a mean age of 49 years (table 1).

### Uptake of lateral flow testing

Since the introduction of guidance recommending asymptomatic testing for all adults two times per week, 16.9% (95% CI 16.0% to 17.8%, n=1123/6646) of people reported that they had taken a lateral flow test for

**Table 1** Participant characteristics

| Participant characteristic | Level | N | Percentage |
|---|---|---|---|
| Sex | Male | 3025 | 45.5 |
| | Female | 3600 | 54.2 |
| | Other / prefer not to say | 21 | 0.3 |
| Age | Range 18 to 91 years | Mean = 49.4 years | SD = 16.6 |
| Region | East Midlands | 539 | 8.1 |
| | East of England | 713 | 10.7 |
| | London | 815 | 12.3 |
| | North East | 312 | 4.7 |
| | North West | 815 | 12.3 |
| | Scotland | 685 | 10.3 |
| | South East | 924 | 13.9 |
| | South West | 604 | 9.1 |
| | West Midlands | 627 | 9.4 |
| | Yorkshire and the Humber | 612 | 9.2 |
| Ethnicity | White British | 5590 | 84.1 |
| | White other | 397 | 6 |
| | Mixed | 137 | 2.1 |
| | Asian / Asian British | 305 | 4.6 |
| | Black / Black British | 142 | 2.1 |
| | Arab / other | 34 | 0.5 |
| | Prefer not to say | 41 | 0.6 |
| Highest earner in household works in manual occupation | No | 4688 | 70.5 |
| | Yes | 1812 | 27.3 |
| | Prefer not to say | 146 | 2.2 |
| Index of Multiple Deprivation | 1st quartile (least deprived) | 1365 | 20.5 |
| | 2nd quartile | 1652 | 24.9 |
| | 3rd quartile | 1791 | 26.9 |
| | 4th quartile (most deprived) | 1838 | 27.7 |

**Table 2** Uptake of lateral flow testing

| When was the last time you had a test for coronavirus? We are interested in your most recent test, even if you did not have symptoms (total n=6646) | % (n) | Asked to people who reported having a COVID-19 test in the last 7 days. And how many times have you taken a test for COVID-19 in the last 7 days? (total n=1123) | % (n) |
|---|---|---|---|
| Within the last 24 hours | 4.1 (273) | Once | 34.3 (385) |
| 1–3 days ago | 7.4 (492) | 2 times | 47.2 (530) |
| 4–7 days ago | 5.4 (358) | 3 times | 9.4 (106) |
| 1–2 weeks ago | 7.1 (469) | 4–5 times | 3.7 (42) |
| 2–4 weeks ago | 6.9 (458) | 6–7 times | 2.8 (32) |
| 1–3 months ago | 10.0 (666) | 8–9 times | 0.7 (8) |
| 3–6 months ago | 7.8 (519) | 10 times or more | 1.2 (14) |
| More than 6 months ago | 6.3 (417) | | |
| I have never had a COVID-19 test | 43.0 (2861) | | |
| Do not know | 2.0 (133) | Do not know | 0.5 (6) |

**Table 3** Where people register the results of their latest test

| Asked to people who reported having a COVID-19 test in the last 7 days. How, if at all, did you report the result of your test? Tick all that apply (total n=1123) | % (n) |
|---|---|
| I registered my result on GOV.UK | 50.0 (561) |
| I registered my result by phone with NHS Test and Trace | 17.0 (191) |
| I informed my employer | 15.4 (173) |
| I informed the school, nursery or further education college where I or a member of my family study | 6.9 (77) |
| I informed friends/family I was planning to meet after taking the test | 6.3 (71) |
| I informed friends/family I had recently met before taking the test | 5.0 (56) |
| I informed the university where I or a member of my family study | 2.5 (28) |
| Other | 2.0 (23) |
| I did not report the result to anyone | 15.8 (177) |
| Registered test result with GOV.UK or NHS Test and Trace | 64.1 (720) |
| Registered result with anyone (GOV.UK, NHS Test and Trace, one's employer or the school, nursery, further education college or university where the participant or a member of their family study) | 77.2 (867) |

COVID-19 in the last week, excluding those whose most recent test was a PCR test (table 2). Of these, 65.2% had completed two or more tests in the last week (11.0% total sample).

Most people reported the result of their most recent test to someone, with 64% reporting that they registered it with an official government agency (table 3).

### Associations with increased uptake of lateral flow testing
Increased uptake of lateral flow testing was associated with: being female, younger age, having a dependent child in your household, being employed, being vaccinated, having experienced COVID-19 symptoms in the last 7–10 days, being a student, having been out to work in the last week, working in a sector that adopted lateral flow testing early (health or social care, education and childcare, and travel), having heard more about regular lateral flow testing, knowing that you are eligible for regular lateral flow testing, being confident that LFTs are accurate, agreeing that regularly testing people without symptoms is an effective way to prevent the spread of COVID-19 and perceiving a greater risk of COVID-19 to people in the UK (table 4).

Not having had a test was associated with not knowing that you were eligible for regular lateral flow testing, agreeing that you only need to take an LFT if you have come into contact with somebody who has COVID-19 and that people who have been vaccinated do not need to be tested for COVID-19 regularly (in people who reported at least one dose of the vaccine; table 4). There

was significant variation by region, with Scotland showing lower uptake of lateral flow testing.

We ran a sensitivity analysis, conducting logistic regression analyses using index of multiple deprivation quartile as a categorical variable. This made minimal difference to the results.

### DISCUSSION
These data suggest that uptake of lateral flow testing is low, with approximately 17% of the sample reporting having taken a test in the last week. Of the total sample, only 11% report completing at least two LFTs in the last week, in line with government recommendations.[4] This is slightly lower than another survey finding that 25% of English and Scottish adults reported taking regular COVID-19 tests (defined as at least once or two times per week; data collected: 29 July 2021), although that was not in a nationally representative sample.[22] These data are not directly comparable with the Liverpool pilot, which reported uptake of testing over the complete duration of the pilot (almost 6 months), rather than uptake of testing per week.[7 8] In the first month of the pilot, 35% of people reported having taken up either an LFT or PCR test.[23] Analyses of tests reported to the UK Government indicate that the number of LFTs registered had steadily declined from approximately 5.7 million LFTs (15–21 April 2021[24]) to around 3.5 million (27 May 2021–2 June 2021).[25] This

**Table 4** Factors associated with having completed at least one COVID-19 test in the last week. Bold values indicate findings significant at p<0.003

| Factor | Level | Had not taken an LFT in the last week, n (%) (total n=5390) | Had taken at least one LFT in the last week, n (%) (total n=1123) | aOR (95% CI) for having taken at least one LFT in the last week * | P value |
|---|---|---|---|---|---|
| Survey round | 19–21 April 2021 (round 48) | 1381 (84.6) | 252 (15.4) | Reference | – |
| | 4–5 May 2021 (round 49) | 1334 (81.3) | 306 (18.7) | 1.13 (0.93 to 1.38) | 0.22 |
| | 17–19 May 2021 (round 50) | 1326 (82.6) | 280 (17.4) | 1.04 (0.85 to 1.27) | 0.69 |
| | 1–2 June 2021 (round 51) | 1349 (82.6) | 285 (17.4) | 0.97 (0.79 to 1.19) | 0.78 |
| | Overall | – | – | $\chi^2(3)$=2.8 | 0.42 |
| Region | East Midlands | 443 (83.6) | 87 (16.4) | Reference | – |
| | East of England | 564 (80.3) | 138 (19.7) | 1.23 (0.90 to 1.68) | 0.19 |
| | London | 638 (80.7) | 153 (19.3) | 1.13 (0.83 to 1.55) | 0.43 |
| | North East | 251 (81.8) | 56 (18.2) | 1.22 (0.83 to 1.80) | 0.31 |
| | North West | 669 (84.2) | 126 (15.8) | 1.02 (0.75 to 1.40) | 0.89 |
| | Scotland | 608 (90.3) | 65 (9.7) | **0.53 (0.37 to 0.76)** | 0.001 |
| | South East | 726 (80.3) | 178 (19.7) | 1.24 (0.92 to 1.67) | 0.15 |
| | South West | 504 (84.3) | 94 (15.7) | 1.04 (0.74 to 1.44) | 0.83 |
| | West Midlands | 489 (79.5) | 126 (20.5) | 1.35 (0.98 to 1.85) | 0.07 |
| | Yorkshire and the Humber | 498 (83.3) | 100 (16.7) | 1.08 (0.78 to 1.50) | 0.65 |
| | Overall | – | – | **$\chi^2(9)$=35.6** | <0.001 |
| Gender | Male | 2508 (85.0) | 442 (15.0) | Reference | – |
| | Female | 2868 (80.9) | 676 (19.1) | **1.32 (1.14 to 1.51)** | <0.001 |
| Age | Raw age | n=5390, M=50.6, SD=16.5 | n=1123, M=44.9, SD=15.9 | **0.76 (0.72 to 0.81)** | <0.001 |
| Age–quadratic (age–mean)$^2$ | – | – | – | 1.0003 (1.0000 to 1.0006) | 0.06 |
| Dependent child in household | None | 3829 (85.3) | 660 (14.7) | Reference | – |
| | Child present | 1561 (77.1) | 463 (22.9) | **1.29 (1.10 to 1.51)** | 0.001 |
| Has a chronic illness (oneself) | None | 3974 (82.2) | 819 (17.8) | Reference | – |
| | Present | 1478 (83.6) | 209 (16.4) | 1.18 (1.00 to 1.39) | 0.05 |
| Household member has chronic illness | None | 4482 (82.7) | 936 (17.3) | Reference | – |
| | Present | 790 (82.0) | 173 (18.0) | 1.04 (0.86 to 1.26) | 0.71 |
| Employment status | Not working | 2544 (88.5) | 330 (11.5) | Reference | – |
| | Working | 2783 (78.1) | 781 (21.9) | **1.94 (1.63 to 2.32)** | <0.001 |
| Highest earner in household works in manual occupation | No | 3844 (83.6) | 754 (16.4) | Reference | – |
| | Yes | 1438 (80.5) | 348 (19.5) | 1.10 (0.94 to 1.29) | 0.21 |
| Index of multiple deprivation† | 1st quartile (least deprived) | 1125 (83.7) | 219 (16.3) | 0.93 (0.87 to 0.99) | 0.02 |
| | 2nd quartile | 1319 (80.8) | 313 (19.2) | | |
| | 3rd quartile | 1439 (82.1) | 314 (17.9) | | |
| | 4th quartile (most deprived) | 1507 (84.5) | 277 (15.5) | | |
| Highest educational or professional qualification | GCSE/vocational/A level/no formal qualifications | 3625 (82.9) | 749 (17.1) | Reference | – |
| | Degree or higher (bachelors, masters and PhD) | 1765 (82.5) | 374 (17.5) | 0.91 (0.78 to 1.07) | 0.24 |
| Ethnicity | White British | 4549 (82.7) | 954 (17.3) | Reference | – |
| | White other | 323 (83.9) | 62 (16.1) | 0.87 (0.61 to 1.25) | 0.45 |
| | Black and minority ethnicity | 484 (82.2) | 105 (17.8) | 0.83 (0.64 to 1.08) | 0.16 |
| | Overall | – | – | $\chi^2(2)$=2.1 | 0.34 |
| English as a first language | No | 415 (83.2) | 84 (16.8) | Reference | – |
| | Yes | 4975 (82.7) | 1039 (17.3) | 1.19 (0.86 to 1.65) | 0.29 |
| Had COVID-19 before | Think not | 4600 (83.8) | 888 (16.2) | Reference | – |
| | Think yes | 790 (77.1) | 235 (22.9) | 1.23 (1.03 to 1.47) | 0.02 |

Continued

**Table 4** Continued

| Factor | Level | Had not taken an LFT in the last week, n (%) (total n=5390) | Had taken at least one LFT in the last week, n (%) (total n=1123) | aOR (95% CI) for having taken at least one LFT in the last week * | P value |
|---|---|---|---|---|---|
| Vaccination status | Not vaccinated | 1628 (82.9) | 336 (17.1) | Reference | – |
| | 1 dose | 1937 (83.2) | 392 (16.8) | **1.52 (1.25 to 1.86)** | <0.001 |
| | 2 doses | 1825 (82.2) | 395 (17.8) | **2.45 (1.96 to 3.07)** | <0.001 |
| | Overall | – | – | $\chi^2$(2)=61.7 | <0.001 |
| Financial hardship | Range: 3 (least)–15 (most) | N=5311, M=7.3, SD=3.0 | N=1107, M=7.5, SD=3.0 | 0.99 (0.97 to 1.01) | 0.38 |
| COVID-19 symptoms in last week/10 days | No | 5258 (83.2) | 1061 (16.8) | Reference | – |
| | Yes | 132 (68.0) | 62 (32.0) | **1.89 (1.34 to 2.66)** | <0.001 |
| Being a student | No | 5189 (83.0) | 1061 (17.0) | Reference | – |
| | Yes | 138 (73.4) | 50 (26.6) | **2.65 (1.76 to 4.00)** | <0.001 |
| Been out to work in last week | No | 3702 (88.3) | 490 (11.7) | Reference | – |
| | Yes | 1688 (72.7) | 633 (27.3) | **2.30 (1.94 to 2.73)** | <0.001 |
| Number of times been out to meet people from another household socially | Range: 0–30 | N=5390, M=0.9, SD=1.5, median=0 | N=1123, M=1.2, SD=1.6, median=1 | 1.05 (1.01 to 1.10) | 0.03 |
| Work in a sector that adopted LFT early | No | 4700 (86.3) | 744 (13.7) | Reference | – |
| | Yes | 690 (64.5) | 379 (35.5) | **2.54 (2.14 to 3.02)** | <0.001 |
| Amount heard about regular LFT | 4-point scale from 'nothing at all' to 'a great deal' | N=5253, M=2.8, SD=0.8 | N=1112, M=3.3, SD=0.7 | **2.28 (2.06 to 2.51)** | <0.001 |
| As far as you know, are you eligible to receive rapid COVID-19 tests two times per week to check for COVID-19 even if you do not have symptoms (also known as lateral flow testing)? | No | 928 (90.5) | 97 (9.5) | Reference | – |
| | Do not know | 1718 (95.0) | 91 (5.0) | **0.59 (0.43 to 0.80)** | 0.001 |
| | Yes | 2744 (74.6) | 935 (25.4) | **2.98 (2.35 to 3.78)** | <0.001 |
| | Overall | – | – | $\chi^2$(2)=240.5 | <0.001 |
| I am confident that LFTs are accurate | 5-point scale from 'strongly disagree' to 'strongly agree' | N=5131, M=3.3, SD=1.0 | N=1097, M=3.6, SD=0.9 | **1.40 (1.29 to 1.51)** | <0.001 |
| Regularly testing people without symptoms is an effective way to prevent the spread of COVID-19 | 5-point scale from 'strongly disagree' to 'strongly agree' | N=5225, M=3.9, SD=0.9 | N=1115, M=4.3, SD=0.8 | **1.96 (1.77 to 2.16)** | <0.001 |
| I do not need to take an LFT unless I have come into contact with somebody who has COVID-19 | 5-point scale from 'strongly disagree' to 'strongly agree' | N=5061, M=2.6, SD=1.0 | N=1114, M=2.0, SD=1.1 | **0.51 (0.47 to 0.55)** | <0.001 |
| People who have been vaccinated do not need to be tested for COVID-19 regularly | 5-point scale from 'strongly disagree' to 'strongly agree' | | | | |
| | In people who have not been vaccinated | N=1480, M=2.8, SD=1.1 | N=329, M=2.7, SD=1.1 | 0.90 (0.80 to 1.01) | 0.08 |
| | In people who have had one vaccine dose | N=1790, M=2.5, SD=1.0 | N=385, M=2.0, SD=1.1 | **0.54 (0.47 to 0.61)** | <0.001 |
| | In people who have had two vaccine doses | N=1644, M=2.6, SD=1.0 | N=392, M=2.0, SD=1.1 | **0.53 (0.47 to 0.60)** | <0.001 |
| Perceived risk of COVID-19 to self | 5-point scale from 'no risk at all' to 'major risk' | N=5343, M=3.0, SD=1.1 | N=1117, M=3.0, SD=1.1 | 1.04 (0.98 to 1.11) | 0.23 |
| | Excluding people who tested positive and whose test result was inconclusive | N=5343, M=3.0, SD=1.1 | N=1058, M=3.0, SD=1.1 | 1.03 (0.96 to 1.10) | 0.42 |
| Perceived risk of COVID-19 to people in the UK | 5-point scale from 'no risk at all' to 'major risk' | N=5326, M=3.5, SD=1.0 | N=1114, M=3.6, SD=0.9 | **1.13 (1.05 to 1.22)** | 0.001 |

*Adjusting for survey round, region, gender, age (raw and quadratic), the presence of a dependent child in the household, having a chronic illness oneself, having a household member who has chronic illness, employment status, highest earner in household works in manual occupation, index of multiple deprivation (continuous variable), highest educational or professional qualification, ethnicity, first language, having had COVID-19 before, vaccination status and financial hardship.
†Treated as a continuous variable.
aOR, adjusted odds ratio; LFT, lateral flow test.

number includes tests taken by children and does not include tests that have not been officially registered on the UK Government website. However, our data would imply around 10 million LFTs should be reported each week by people aged over 17 years in England and Scotland alone, suggesting that our survey respondents may

be more compliant (uptake of testing and/or reporting of testing) than the general population. This is corroborated by official figures estimating that approximately 21% of LFTs are reported.[26] Our data indicate that 64% of participants' most recent tests had been registered with an official government agency.

Factors related to employment or study were associated with uptake of lateral flow testing. People were more likely to report having a test in the last week if they were employed, had been out to work in the last week, and if they worked in a sector that recommended asymptomatic testing before the national guidance was implemented. Students were also more likely to report having tested in the last week. This could be because people were encouraged or compelled to take tests through their workplace, because they were more familiar with testing, or because they were more worried or perceived a greater risk of exposure to COVID-19 as they were going out to their place of work or study.[27] The current findings suggest that encouraging employees to take tests could drive uptake. However, this should be approached with caution. Qualitative research suggests that barriers to implementing testing in the workplace include perceived inaccuracy of LFTs and adding to employee burden.[28] There are also ethical issues to consider in employers putting pressure on their employees, and there is a potential resulting lack of income if workers are unable to attend their place of work if they decline.[29] Mandating testing may result in negative attitudes toward testing becoming more entrenched.[30] Uptake could also be increased by making testing easier, for example, at or very near to places of work or study, drop-in rather than appointment based and with explicit paid time off for testing. From 4 October 2021, people must request a 'collect code' online or by telephone in order to collect a packet of LFTs from pharmacies.[31] Previously, this was not needed. How this has impacted uptake of LFTs is not yet clear, but it is likely that this change will make LFTs less accessible to some and has the potential to negatively impact uptake.

Uptake of lateral flow testing was higher in people who reported experiencing cardinal COVID-19 symptoms (high temperature, a new, continuous cough, a loss or change to sense of smell or taste) in the last week. UK Government recommendations state that people with cardinal COVID-19 symptoms should request a PCR test, rather than rely on an LFT. It is clear that this requirement is not always being followed.[10] Research suggests that people with less severe SARS-CoV-2 manifest different symptoms at the start of infection.[32] Expanding the symptom set for eligibility for a PCR test is likely to increase testing burden on NHS Test and Trace. In England, all legal restrictions on social mixing were lifted on 19 July 2021.[33] Since this date, a 7-day average of over 20 000 new COVID-19 cases per day has been recorded,[34] with 527 077 (21 August 2021)–1 175 617 (6 September

2021) pillar 2 tests being conducted per day (data checked until 24 November 2021).[35] Official communications aiming to promote engagement with Government recommendations should emphasise that people with COVID-19 symptoms should request a PCR test, as should those who test positive using an LFT.

Socio-demographic factors associated with uptake of lateral flow testing included being younger and living with a dependent child. Previous research has found these factors to be consistently associated with non-adherence to behaviours that prevent the spread of COVID-19.[11 18 36] However, the association between increased lateral flow testing and lower age has also been found in other data, largely driven by those of working age being more likely to complete a test.[8 22] One possible explanation may be that younger people are less likely to work from home.[29 37] Therefore, these findings may be an artefact of people testing in relation to their work or study. Official figures of registered tests indicate that asymptomatic testing in school-aged children, who are 'expected to test twice weekly'[16] under the supervision of their parents, is driving uptake, with numbers of tests conducted falling during the school holidays.[25] Parents may be likely to test themselves for COVID-19 while supervising their child's test. Increased uptake of testing was associated with having been vaccinated. This may reflect general adherence, with those being more likely to engage in preventive behaviours also being more likely to be vaccinated (itself a preventive behaviour).

These data indicate that people were more likely to engage in lateral flow testing if they had heard more about, and knew they were eligible for, regular LFTs. This is consistent with uptake of preventive behaviours in previous pandemics.[38] In line with predictions from the protection motivation theory,[39] perceiving testing to be more accurate and effective was associated with increased uptake.[40] Conversely, people who agreed that you only need to take a test if you have come into contact with a COVID-19 case, and that people who have been vaccinated do not need to be tested regularly, were less likely to have taken a test in the last week. The latter belief was particularly strongly associated with low uptake in those who had been vaccinated. Taken together, these results suggest that media campaigns raising awareness that all adults are eligible for the mass asymptomatic testing programme are likely to increase uptake.

Testing alone cannot prevent transmission of disease. Only when used in conjunction with other behavioural interventions (eg, staying at home and isolating) will testing prevent transmission. Due to its consistent use in combination with other interventions, it is difficult to determine the effectiveness of asymptomatic testing programmes alone.[2 3] One study suggests that mass testing 5% of the UK population per week would lead to a 2% mean reduction in the reproductive rate of SARS-CoV-2.[41] However, this was before the implementation of vaccination. Test sensitivity and specificity, and

prevalence of infection in the population, will affect the number of cases accurately identified and missed by tests.[42] Despite initial concern over the sensitivity of LFTs, especially in asymptomatic cases where people are testing themselves,[43] recent data suggest that LFTs have an absolute sensitivity of over 80% to detect individuals shedding SARS-CoV-2.[44]

We cannot be certain that the behaviour and beliefs of those that complete internet surveys are representative of those of the general population. This is reflected in the higher reporting of LFTs in our sample compared with that reported by official agencies. However, associations within the data are still likely to be informative.[45] Since data reflected self-reported behaviour, reports may be biased and influenced by social desirability or poor recall. Given that we asked about behaviour in the past week, the influence of poor recall should be low. We also mitigated this by defining uptake as having completed one test in the last week, while Government guidelines suggest two COVID-19 tests per week should be completed. Although we have data on where participants received their LFTs, we did not ask why they took their most recent tests and so cannot identify whether people are engaging in routine testing, or whether testing behaviour is driven by completing a test before socialising or attending work.

When used in tandem with self-isolation, testing can prevent the spread of COVID-19 by lowering the circulation of cases within the community. In the UK, recommendations at the time of data collection stated that people with cardinal symptoms should complete a PCR test alongside a two times per week asymptomatic mass testing programme. People who tested positive using an LFT should then complete a PCR test. Mass testing programmes aim to identify cases in the population that may otherwise have been missed. However, their effectiveness is unclear. Our study suggests that uptake of lateral flow testing in the population is low. Interventions to prevent the spread of COVID-19 are unlikely to be effective if people do not engage with the behaviour. One reason for low uptake is that people do not know they are eligible for regular asymptomatic testing. Work-related and study-related factors were associated with uptake of lateral flow testing. Encouragement of employees and students, especially those attending their place of work or study, to engage in asymptomatic testing may increase uptake. However, employers and educational institutions should exercise caution so as not to place undue pressure on employees and students to test. People with symptoms were more likely to have completed a test in the past week. Consideration should be given to how best to optimise testing in the UK (PCR and LFT), taking into account the full range of symptoms displayed early in SARS-CoV-2 infection, transmission when asymptomatic, financial cost of testing and burden on the testing system. Communications aiming to promote engagement with Government-recommended testing should highlight that people with cardinal COVID-19 symptoms should request a PCR test rather than take an LFT.

**Author affiliations**
[1]Institute of Psychiatry, Psychology and Neuroscience, King's College London, London, UK
[2]NIHR Health Protection Research Unit in Emergency Preparedness and Response, London, UK
[3]Institute of Health Informatics, University College London, London, UK
[4]UK Health Security Agency, Salisbury, UK
[5]King's Centre for Military Health Research and Academic Department of Military Mental Health, King's College London, London, UK
[6]Centre for Behaviour Change, University College London, London, UK

**Contributors** All authors conceptualised the study and contributed to survey materials. LES completed analyses with guidance from HWWP and GJR. LES wrote the first draft of the manuscript. HWWP, RA, NTF, SM and GJR contributed to subsequent drafts of the manuscript. LES, HWWP, RA, NTF, SM and GJR approved the final manuscript. GJR is the guarantor. The corresponding author attests that all listed authors meet authorship criteria and that no others meeting the criteria have been omitted.

**Funding** This work was funded by the National Institute for Health Research (NIHR) Health Services and Delivery Research programme (project reference number: 11/46/21). Surveys were commissioned and funded by Department of Health and Social Care, with the authors providing advice on the question design and selection. LES, RA and GJR are supported by the NIHR Health Protection Research Unit (HPRU) in Emergency Preparedness and Response, a partnership between the UK Health Security Agency, King's College London, and the University of East Anglia. RA is also supported by the NIHR HPRU in Behavioural Science and Evaluation, a partnership between the UK Health Security Agency and the University of Bristol. HWWP has received funding from Public Health England and NHS England. NTF is partly funded by a grant from the UK Ministry of Defence. The views expressed are those of the authors and not necessarily those of the NIHR, UK Health Security Agency, the Department of Health and Social Care or the Ministry of Defence. The Department of Health and Social Care funded data collection (no grant number).

**Competing interests** All authors have completed the ICMJE uniform disclosure form at www.icmje.org/coi_disclosure.pdf and declare: all authors had financial support from NIHR for the submitted work; RA is an employee of the UK Health Security Agency; HWWP has received additional salary support from Public Health England and NHS England, received consultancy fees to his employer from Ipsos MORI and has a PhD student who works at and has fees paid by Astra Zeneca; no other financial relationships with any organisations that might have an interest in the submitted work in the previous 3 years; and no other relationships or activities that could appear to have influenced the submitted work. NTF is a participant of an independent group advising NHS Digital on the release of patient data. All authors are participants of the UK's Scientific Advisory Group for Emergencies or its subgroups.

**Patient and public involvement** Patients and/or the public were not involved in the design, or conduct, or reporting, or dissemination plans of this research.

**Patient consent for publication** Not applicable.

**Ethics approval** This study involves human participants. This work was conducted as a service evaluation of the Department of Health and Social Care's public communications campaign and, following advice from King's College London Research Ethics Subcommittee, was exempt from ethical approval. Participants were recruited from a panel of people who had signed up to take part in online surveys. Completing the survey implied consent. This is a standard practice in market research surveys.

**Provenance and peer review** Not commissioned; externally peer reviewed.

**Data availability statement** No data are available. No additional data are available from the authors.

**ORCID iDs**
Louise E Smith http://orcid.org/0000-0002-1277-2564
Henry WW Potts http://orcid.org/0000-0002-6200-8804
Richard Amlôt http://orcid.org/0000-0003-3481-6588
Nicola T Fear http://orcid.org/0000-0002-5792-2925
Susan Michie http://orcid.org/0000-0003-0063-6378
G James Rubin http://orcid.org/0000-0002-4440-0570

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
