## [Reviewer comments · BMJ Open]

ARTICLE DETAILS

TITLE (PROVISIONAL)	Who is engaging with lateral flow testing for COVID-19 in the UK? The COVID-19 Rapid Survey Of Adherence To Interventions And Responses (CORSAIR) study
AUTHORS	Smith, Louise; Potts, Henry; Amlot, Richard; Fear, Nicola; Michie, Susan; Rubin, GJ

VERSION 1 – REVIEW

REVIEWER	Green, Mark University of Liverpool, Geography & Planning
REVIEW RETURNED	15-Oct-2021

GENERAL COMMENTS	Thank you to the authors for their hard work in writing this paper. I enjoyed reading the paper and it is clear that lots of effort has been put into the manuscript. I would be happy to recommend that the paper is accepted for publication, following some minor revisions. What was good - Writing style is good with very few errors. Clear, concise and well written.- Covers main expected areas and offers reasonable explanations for study results.- The methodology was clear throughout and easy to see what you did. Inclusion of PPI elements in the early parts of the design was nice to see.- Concise results section - I like this as sharp and to the point. What could be improved Strengths and limitations Bullet points include both that the surveys are both representative (first point) and not representative (last point). The data are one or the other surely. Suggest removing one bullet point for consistency in messages. The first bullet point says English population, however surveyed participants were England and Scotland. Suggest revise. Introduction “Testing was likely driven by the intense media campaign in Liverpool, and the novelty of the testing at the time” Would explain uptake during the initial pilot, but does not explain experiences thereafter and is a little reductionist of the reasons why people got tested from January onwards. Suggest revise to “Testing in the initial months was partly driven by the intense media campaign in Liverpool, and the novelty of the testing at the time.”
--

The Liverpool 'mass testing' pilot reports are cited and it would also be useful to reference the associated Lancet Regional Health - Europe paper given it was focused on inequalities too. I am also not a fan of the whole 'cite my work' element of reviews, so feel free to tell me off, but given the paper itself was focused on inequalities in uptake (as opposed to the longer report which was less detailed in that regard) it does form a helpful citation for several parts of the paper (here and other sections). I'll leave it with you to decide.

Green et al. 2021. Evaluating social and spatial inequalities of large scale rapid lateral flow SARS-CoV-2 antigen testing in COVID-19 management: An observational study of Liverpool, UK (November 2020 to January 2021). The Lancet Regional Health - Europe 6: 100107. <https://doi.org/10.1016/j.lanep.2021.100107>.

Methodology

"Quotas were applied based on age and gender (combined) to ensure the sample was broadly representative of the population." Quota sampling does not ensure representativeness. I would revise this point to something like "Quotas were applied based on age and gender (combined) to try to match the English and Scottish population structure." - assuming this is what you did.

Index of multiple deprivation was derived from postcode. Please clarify which index was used (I assume was 2019 version)

Whether people had COVID-19 feels better placed in the contextual section. It is not a socio-demographic factor.

While the study was service evaluation and exempt from ethical approval, it would be nice that there is a statement that the data were collected with consent.

I would remove the power calculation statement. If your goal is to understand the determinants of lateral flow testing, then that is the purpose of your analyse rather than to estimate some prevalence (which isn't really what you did). It doesn't really fit in well here or add anything.

I am slightly concerned by the sheer number of covariates include in the analysis and possible issues this brings in the modelling process (so called 'Table 2 errors'). It is not my role to make you pick apart of all this mind you, but I wondered whether you had time to reflect over the stability of associations and could comment on this issue. Bonferroni correction is one option, but doesn't necessary help always.

I was unclear how index of multiple deprivation was included in the model. In Table 4 the factor is described as 1st quartile to 4th quartile (i.e., categorical), but only a single coefficient is provided suggesting that it was treated as a continuous variable. This is incorrect and may produce misleading results. Suggest either (i) replace factor with index score and re-run analysis as continuous, or (ii) re-run as categorical with 1st quartile as reference.

Discussion

The sentence "Factors related to employment or study were associated with uptake of lateral flow testing, with people being more likely to report having a test in the last week if they were

	employed, had been out to work in the last week, if they worked in a sector that recommended asymptomatic testing before the national guidance was implemented, and if they were a student” Is a bit too long and hard to follow - suggest cutting it up. Short and simple always. The age association (p16) is an interesting finding and matches what appears across the wider literature. It also is partly explained by low uptake in the older adults (e.g., age 60+) rather than just behaviours in younger adults. There might be a generational divide in testing behaviours, that could be where we need to improve our messaging to. It might be useful to reflect on this too. The statement “Although socio-demographic characteristics of the sample were broadly reflective of the UK population” is not actually supported with any data though. We do not know what your study population looks like. Might be useful to have a summary table for age, sex, region, SES in the results and some reflection over how representative it is. It is slightly different to the data in Table 4. This is an important point about how generalisable your findings are, which is unclear from the data presented. For example, it looks from Table 4 that your data are older than the population average, with more females (55.6%) than expected - I needed to calculate the latter myself so a table that can be summarised as ‘not quite representative, but also not too off’ would be useful. If the data are not quite representative, then the text throughout needs to be edited to not make such a claim.
--	---

REVIEWER	Nguyen, Le Khanh Ngan Strathclyde Business School Department of Management Science
REVIEW RETURNED	24-Oct-2021

GENERAL COMMENTS	In this study, the authors adopted cross-sectional surveys to investigate the uptake of lateral flow testing. The paper is well-written and provides a snapshot of the factors associated with uptake of lateral flow testing. More detailed comments are as below: 1/ The paper offers a good illustration of the characteristics of people who adhere to the testing policy - but does not provide insights into the underlying reasons/barriers for low adherence. Have the authors considered the appropriateness of the study methods and justified the methods? 2/ Quota sampling is generally not as representative as biomedical research surveys. Therefore, generalisation from such surveys is doubted. 3/ It is not clear how the authors choose the psychological, contextual, and socio-demographic factors to consider in the study. 4/ The results are reported based on data collected in a series of four surveys. It would be interesting to know whether the results between surveys are consistent and in line with the overall results across four surveys. As the perceived risk of increased SARS-CoV-2 prevalence in the community may affect the uptake of testing, it would be also interesting to investigate such association. 5/ The results reported in the study are generally predictable. Implications of the study results upon policy are mentioned but not clear. 6/ Potential biases have not been well assessed (e.g., age in relation to survey participation rate)
--

	7/ The description on the association between the uptake of testing and thinking that you have had COVID-19 before seems ambiguous. Not sure what the value the results on this would add. Why not examine the association between uptake and having had confirmed COVID-19 positive before? 8/ The relationship between adherence to lateral flow testing and reporting and how that affects the results have not been addressed.
--	--

REVIEWER	Crozier, Alex University College London, Division of Biosciences
REVIEW RETURNED	07-Nov-2021

GENERAL COMMENTS	Dear Editor, This manuscript contains some important data points which contribute to the existing evidence on LFT uptake, and reasons behind differences in uptake, and perceptions around the mass testing underway in UK. This evidence is relevant to monitoring and improving the effectiveness of the currently very poorly implemented and wasteful LFT programme and PCR testing system. While I would broadly support the publication of this article for the reasons above, I believe the comments I have made below should be addressed before publication. Page 6 line 6-10 Would be clearer on definitions of asymptomatic infection and transmission – much of literature muddles this and important to be clear, when talking about testing programmes and uptake, etc, that definitions are precise so as to not further muddy the literature here. See papers here for justification for this: https://www.sciencedirect.com/science/article/pii/S1473309920308379 https://www.bmj.com/content/374/bmj.n1625 “Transmission while asymptomatic” may be more appropriate here over “asymptomatic cases account for x% of cases.” Page 6 line 17 Reference 3 is no longer just pre-print, and has been reviewed and published: https://pubmed.ncbi.nlm.nih.gov/33758017/ I would also consider adding in a reference to the full Liverpool community testing report published on July 7, rather than just a commentary of it as reference 2 is. Page 6 line 40-42 See first point about importance of symptom definitions – I would argue “key COVID” symptoms is slightly misleading – many were (officially) ineligible for a PCR test as they did not present with ‘official’ UK gov symptoms, but as headache, fatigue etc are as (or more) common and earlier presenting than eg cough and fever, defining some as “key” and therefore implying others not “key” is questionable. The symptoms most relevant for early identification of cases, and therefore for triaging for testing to exert transmission control are those which are currently not listed on any government or NHS websites. https://www.sciencedirect.com/science/article/pii/S1473309920308379 https://www.bmj.com/content/374/bmj.n1625
---

Page 7 lines 36-38

Unsure if waves is the best terms here (and in below text), given context of discussion is to mitigate epidemic 'waves..' – possibly better to use 'rounds'?

Page 11 lines 35-42

Is there any data here on how many of the 16.9% took tests as part of asymptomatic testing/screening, rather than because of experiencing symptoms? As this is the key question of this paper here – how many people participating in asymptomatic testing programme – rather than how many people took a lateral flow test in the last week - which cannot be answered without data on symptomatic vs asymptomatic LFT testing.

Table 2

Was there data to report on difference in reporting testing to Gov/friends&family if tests negative or positive..?

Page 16 line 9

I think worth reconfirming, and discussing implications of, here also that, of the 17% who took a test in the last week, 65% had taken 2 in the last week (11% of total sample). This suggests even lower than 17% adherence to the policy of biweekly testing (in terms of asymptomatic testing, regular (twice/thrice weekly not one-off)) or pre-event/socialising testing is key to lateral flow tests being effective in reducing transmission, not just a random LFT every now and then.. Important to try and distinguish between these: A pre-event, or pre-socialising Friday night test has value (imo), but is different from biweekly testing, and I would suspect engagement and adherence to such testing different, also.

Pre event/pre-socialising testing, and early testing when experiencing symptoms, may be a more realistic policy option which should be considered, and may be a way to increase uptake (and increase benefits realised), rather than asking the public to engage with biweekly testing for months on end even though they have been vaccinated.

Page 16 line 33-36

Here again one reason for discrepancy here may be because people more likely to report positive, rather than negative, LFT results. As don't know proportion testing genuinely as part of regular asymptomatic regular testing, or just because they developed (mild or 'official') symptoms, difficult to interpret reasons for differences between data sources here.

Page 17 line 4-6

RE difficulties in adherence and challenges of implementation even when required by employers, I would include at least one of these valuable references from Liverpool pilot:

<https://academic.oup.com/ageing/advance-article/doi/10.1093/ageing/afab162/6322881>

<https://bmchealthservres.biomedcentral.com/articles/10.1186/s12913-021-07191-9>

Page 17 line 10

Disagree with the 'people with symptoms should request a PCR rather than rely on LFT.'

	If everyone with relevant symptoms (including symptoms most relevant for identification early enough for transmission control such as headache and fatigue) requested a PCR the system would immediately be swamped. Government policy is that those with symptoms should request a PCR, but this does not mean this is the most optimal way for the system to work (before even get on to cost-effectiveness of continued mass PCR testing when little else being done to control transmission..), and this is a crucial distinction to make, imo. Authors shouldn't assume those with symptoms 'should' seek a PCR when it is unclear if this is really best public health option. Concluding remarks In concluding remarks I think could expand a little more, to include that communications on which test to take and when (and why), and how to act after receipt of a result in a given context, should be far clearer at this stage of the epidemic, and also that testing will never be particularly effective in curbing transmission when there is so little focus on the end to end process (symptom recognition through to action after receipt of the result). Perhaps more could also be said as to whether biweekly testing in a largely vaccinated population is still a proportionate response, especially as uptake is so low to have near negligible effect, and whether this is a distraction from what really matters – should greater focus and resources be given to identify and isolate symptomatic cases (full relevant symptom range) early, and also increase engagement with pre-event/pre-socialising testing. If system optimised towards these individuals where impact on epidemic greatest, rather than incredibly poorly implemented mass screening of low risk populations, benefits likely to outweigh biweekly testing with 11% uptake.. Alex Crozier
--	---

VERSION 1 – AUTHOR RESPONSE

Reviewer: 1

Dr. Mark Green, University of Liverpool

Comments to the Author:

Thank you to the authors for their hard work in writing this paper. I enjoyed reading the paper and it is clear that lots of effort has been put into the manuscript. I would be happy to recommend that the paper is accepted for publication, following some minor revisions.

What was good

- Writing style is good with very few errors. Clear, concise and well written.
- Covers main expected areas and offers reasonable explanations for study results.
- The methodology was clear throughout and easy to see what you did. Inclusion of PPI elements in the early parts of the design was nice to see.
- Concise results section - I like this as sharp and to the point.

We thank the reviewer for taking the time to read and comment on our manuscript.

What could be improved

Strengths and limitations

Bullet points include both that the surveys are both representative (first point) and not representative (last point). The data are one or the other surely. Suggest removing one bullet point for consistency in messages.

We have removed the words “nationally representative” from the first bullet point.

The first bullet point says English population, however surveyed participants were England and Scotland. Suggest revise.

We have revised this so that it the first bullet point now states “the English and Scottish population”.

Introduction

“Testing was likely driven by the intense media campaign in Liverpool, and the novelty of the testing at the time” Would explain uptake during the initial pilot, but does not explain experiences thereafter and is a little reductionist of the reasons why people got tested from January onwards. Suggest revise to “Testing in the initial months was partly driven by the intense media campaign in Liverpool, and the novelty of the testing at the time.”

We have made this change.

The Liverpool ‘mass testing’ pilot reports are cited and it would also be useful to reference the associated Lancet Regional Health - Europe paper given it was focused on inequalities too. I am also not a fan of the whole ‘cite my work’ element of reviews, so feel free to tell me off, but given the paper itself was focused on inequalities in uptake (as opposed to the longer report which was less detailed in that regard) it does form a helpful citation for several parts of the paper (here and other sections). I'll leave it with you to decide.

Green et al. 2021. Evaluating social and spatial inequalities of large scale rapid lateral flow SARS-CoV-2 antigen testing in COVID-19 management: An observational study of Liverpool, UK (November 2020 to January 2021). The Lancet Regional Health - Europe 6: 100107.

<https://eur03.safelinks.protection.outlook.com/?url=https%3A%2F%2Fdoi.org%2F10.1016%2Fj.lanep.e.2021.100107&data=04%7C01%7C%7Clouise.e.smith%40kcl.ac.uk%7C45e97c51cb7d44f6553d08d9a396c025%7C8370cf1416f34c16b83c724071654356%7C0%7C0%7C637720690129283111%7CUnknown%7CTWfpbGZsb3d8eyJWljoiMC4wLjAwMDAiLCJQljoiv2luMzliLCJBTiI6lk1haWwiLCJXVCi6Mn0%3D%7C1000&data=arChz%2FhFunlhYC4kXAJV3iKXCeuS%2BIFXIoSCdWGAZ8%3D&reserved=0>

Thank you for bringing this paper to our attention, which we agree is very relevant. We have now cited it in the manuscript.

Methodology

“Quotas were applied based on age and gender (combined) to ensure the sample was broadly representative of the population.” Quota sampling does not ensure representativeness. I would revise this point to something like “Quotas were applied based on age and gender (combined) to try to match the English and Scottish population structure.” - assuming this is what you did.

We have amended this sentence so it now reads that quotas were implemented to “try to match these characteristics to those of the general UK population”.

Index of multiple deprivation was derived from postcode. Please clarify which index was used (I assume was 2019 version)

We have clarified that the 2019 version of the index was used, and have provided a citation.

Whether people had COVID-19 feels better placed in the contextual section. It is not a socio-demographic factor.

While we accept that having had COVID-19 before is not technically a socio-demographic factor, it is something that has been significantly associated with our behavioural outcomes in the past, and we wanted to control for it in our analyses, hence we included it in this section. This is the same approach we have taken with other papers.

While the study was service evaluation and exempt from ethical approval, it would be nice that there is a statement that the data were collected with consent.

We have added a sentence stating that, in line with industry standards, consent was implied by participants' completion of the survey.

I would remove the power calculation statement. If your goal is to understand the determinants of lateral flow testing, then that is the purpose of your analyse rather than to estimate some prevalence (which isn't really what you did). It doesn't really fit in well here or add anything.

Rather than removing the power calculation altogether, we have updated the power calculation so that it is now more relevant for the analyses conducted in the manuscript (post-hoc calculation of power achieved to detect small odds ratios in the sample).

I am slightly concerned by the sheer number of covariates include in the analysis and possible issues this brings in the modelling process (so called 'Table 2 errors'). It is not my role to make you pick apart of all this mind you, but I wondered whether you had time to reflect over the stability of associations and could comment on this issue. Bonferroni correction is one option, but doesn't necessary help always.

We conducted a sensitivity analysis in which we conducted logistic regression analyses on socio-demographic factors excluding the variable with the strongest association with uptake of an LFT (vaccine uptake). This had minimal effect on results. Therefore, we believe that our associations are stable, despite the large number of variables included in the logistic regression model.

One of the main criticisms of the Bonferroni correction is that it may be overly conservative. This is particularly relevant for public health research, where small increases in uptake of protective behaviours may have a large effect on health protection when scaled up to the general population. Therefore, by using a Bonferroni correction, we have erred on the side of caution about our interpretation of factors that may be associated with uptake of LFTs. We have given raw p-values, odds ratios, and confidence intervals so that the reader can choose to apply their own adjustments.

I was unclear how index of multiple deprivation was included in the model. In Table 4 the factor is described as 1st quartile to 4th quartile (i.e., categorical), but only a single coefficient is provided suggesting that it was treated as a continuous variable. This is incorrect and may produce misleading results. Suggest either (i) replace factor with index score and re-run analysis as continuous, or (ii) re-run as categorical with 1st quartile as reference.

The reviewer is correct that we included IMD quartiles as a continuous variable in the logistic regression analyses. We are unable to conduct the reviewer's first suggestion as we do not have access to these de-anonymised data. IMD quartile is still clearly an ordinal variable and it is reasonable to treat it as continuous. However, to investigate whether using IMD as a continuous variable produces misleading results, we calculated rates of LFT uptake in each IMD quartile. Confidence intervals for rates in each quartile all overlap and do not suggest a non-linear relationship. Furthermore, we conducted a sensitivity analysis re-running regression analyses using IMD as a categorical variable. This made a minimal difference to results (biggest change in effect size in analyses of socio-demographic variables was 0.01). We have added a note to this effect in the Results section.

Discussion

The sentence "Factors related to employment or study were associated with uptake of lateral flow testing, with people being more likely to report having a test in the last week if they were employed, had been out to work in the last week, if they worked in a sector that recommended asymptomatic testing before the national guidance was implemented, and if they were a student" Is a bit too long and hard to follow - suggest cutting it up. Short and simple always.

We have now split this into three separate sentences, to make this easier to follow.

The age association (p16) is an interesting finding and matches what appears across the wider literature. It also is partly explained by low uptake in the older adults (e.g., age 60+) rather than just behaviours in younger adults. There might be a generational divide in testing behaviours, that could be where we need to improve our messaging to. It might be useful to reflect on this too.

We have added that the association between testing and lower age is likely driven by higher testing rates in people of working age, a pattern of results also seen in the Liverpool pilot (citation added).

The statement "Although socio-demographic characteristics of the sample were broadly reflective of the UK population" is not actually supported with any data though. We do not know what your study population looks like. Might be useful to have a summary table for age, sex, region, SES in the results and some reflection over how representative it is. It is slightly different to the data in Table 4. This is an important point about how generalisable your findings are, which is unclear from the data presented. For example, it looks from Table 4 that your data are older than the population average, with more females (55.6%) than expected - I needed to calculate the latter myself so a table that can be summarised as 'not quite representative, but also not too off' would be useful. If the data are not quite representative, then the text throughout needs to be edited to not make such a claim.

We have included a summary table for participants included in analyses, with reference to age, sex, region, and SES (new table 1). These data show that participants were slightly more likely to be female and white than the general population. This is likely to be because we excluded participants in the overall study from these analyses (excluding people from Wales and Northern Ireland, aged less than 18 years, and whose most recent test was a PCR test or who did not know their most recent test type). In line with this, we have removed references to the sample being "broadly nationally representative" and amended the statement suggested so that it starts "We cannot be certain...".

Reviewer: 2

Dr. Le Khanh Ngan Nguyen, Strathclyde Business School Department of Management Science

Comments to the Author:

In this study, the authors adopted cross-sectional surveys to investigate the uptake of lateral flow testing. The paper is well-written and provides a snapshot of the factors associated with uptake of lateral flow testing.

Thank you for your comments.

More detailed comments are as below:

1/ The paper offers a good illustration of the characteristics of people who adhere to the testing policy - but does not provide insights into the underlying reasons/barriers for low adherence. Have the authors considered the appropriateness of the study methods and justified the methods?

Contextual and psychological factors associated provide insight into underlying reasons and barriers to lateral flow testing. We agree that further work, in particular qualitative research, could be conducted to elicit more detailed reasons for not testing. However, the methods used allow us to quantify the prevalence of self-reported testing and strength of associations between factors and uptake of testing. This would not be possible using qualitative methods. The use of quota sampling allows for rapid data collection, something that is necessary in a fast-moving pandemic situation such as COVID-19.

2/ Quota sampling is generally not as representative as biomedical research surveys. Therefore, generalisation from such surveys is doubted.

We agree that there are biases that arise from the use of quota sampling that may not arise using a random sampling method. However, quota sampling is a pragmatic approach to recruit a large sample over a short space of time. This study is being used to respond to the COVID-19 pandemic, informing communications around behaviours, and a rapid approach is of the essence. We have removed references in the manuscript referring to the sample being “broadly nationally representative”.

3/ It is not clear how the authors choose the psychological, contextual, and socio-demographic factors to consider in the study.

We identified psychological, contextual, and socio-demographic factors to investigate in logistic regression analyses a-priori, based on components of the COM-B model. We have now stated this in the analysis section of the manuscript.

4/ The results are reported based on data collected in a series of four surveys. It would be interesting to know whether the results between surveys are consistent and in line with the overall results across four surveys. As the perceived risk of increased SARS-CoV-2 prevalence in the community may affect the uptake of testing, it would be also interesting to investigate such association.

There was no significant influence of survey wave on uptake. So as not to increase the number of analyses and inflate the possibility of Type 1 and 2 errors, we analysed data from four survey waves together. This increased our power to detect small effect sizes. We have investigated perceived risk of COVID-19 to oneself and people in the UK as part of regression analyses.

5/ The results reported in the study are generally predictable. Implications of the study results upon policy are mentioned but not clear.

We have now expanded the discussion to give further implications with regard to testing policies.

6/ Potential biases have not been well assessed (e.g., age in relation to survey participation rate)

The use of quota sampling means that targets for recruitment into the sample were set based on age and gender (combined). Once quotas are filled, participants of that age/gender profile are prevented from completing the survey. This means that the sample will not be skewed towards people of a younger age, who may be more likely to complete online surveys. In the limitations, we have commented on potential biases that may arise due to the use of a quota sample. Namely, that beliefs and behaviours of people who complete online surveys may not be representative of those in the general population. However, associations within the data are still informative.

7/ The description on the association between the uptake of testing and thinking that you have had COVID-19 before seems ambiguous. Not sure what the value the results on this would add. Why not examine the association between uptake and having had confirmed COVID-19 positive before?

We agree that it would be useful to investigate associations between uptake and having tested positive for COVID-19 before. Our item has been included in the survey since before widespread testing was available, and so was phrased to reflect the situation. However, one of the response options for the item is "I've definitely had it, and had it confirmed by a test". Therefore, we are examining the relationship between uptake of LFTs and having confirmed COVID-19 positive through this variable.

8/ The relationship between adherence to lateral flow testing and reporting and how that affects the results have not been addressed.

The aim of this manuscript was to investigate the prevalence of uptake of lateral flow testing and identify factors associated with uptake. Reporting results of LFTs necessarily takes place after having taken an LFT. Therefore, we are unable to investigate it as an associated factor.

Reviewer: 3

Dr. Alex Crozier, University College London, Queen Mary University of London

Comments to the Author:

Dear Editor,

This manuscript contains some important data points which contribute to the existing evidence on LFT uptake, and reasons behind differences in uptake, and perceptions around the mass testing underway in UK. This evidence is relevant to monitoring and improving the effectiveness of the currently very poorly implemented and wasteful LFT programme and PCR testing system.

While I would broadly support the publication of this article for the reasons above, I believe the comments I have made below should be addressed before publication.

Thank you for your comments.

Page 6 line 6-10

Would be clearer on definitions of asymptomatic infection and transmission – much of literature muddles this and important to be clear, when talking about testing programmes and uptake, etc, that definitions are precise so as to not further muddy the literature here. See papers here for justification for this:

<https://eur03.safelinks.protection.outlook.com/?url=https%3A%2F%2Fwww.sciencedirect.com%2Fscience%2Farticle%2Fpii%2FS1473309920308379&data=04%7C01%7Clouise.e.smith%40kcl.ac.uk%7C45e97c51cb7d44f6553d08d9a396c025%7C8370cf1416f34c16b83c724071654356%7C0%7C0%7C637720690129283111%7CUnknown%7CTWFpbGZsb3d8eyJWljoijoiMC4wLjAwMDAiLCJQljoijoiV2I>

uMzliLCJBTil6Ik1haWwiLCJXVCI6Mn0%3D%7C1000&data=O8SUzh5WnrpyZIKz6vF3tXZLuW
N3NzyUcC0g6DXN%2FaU%3D&reserved=0

<https://eur03.safelinks.protection.outlook.com/?url=https%3A%2F%2Fwww.bmj.com%2Fcontent%2F374%2Fbmj.n1625&data=04%7C01%7Cclouise.e.smith%40kcl.ac.uk%7C45e97c51cb7d44f6553d08d9a396c025%7C8370cf1416f34c16b83c724071654356%7C0%7C0%7C637720690129293099%7CUnknown%7CTWFpbGZsb3d8eyJWIjoiMC4wLjAwMDAiLCJQIjoiV2luMzliLCJBTil6Ik1haWwiLCJXVCI6Mn0%3D%7C1000&data=UO%2Bb8Jl%2FcVEqmpFyDObFfvXfil87L%2B3YIUcVF5cNT0%3D&reserved=0>

“Transmission while asymptomatic” may be more appropriate here over “asymptomatic cases account for x% of cases.”

We thank the reviewer for bringing these papers to our attention and have amended the text in the introduction as suggested.

Page 6 line 17

Reference 3 is no longer just pre-print, and has been reviewed and published:

<https://eur03.safelinks.protection.outlook.com/?url=https%3A%2F%2Fpubmed.ncbi.nlm.nih.gov%2F33758017%2F&data=04%7C01%7Cclouise.e.smith%40kcl.ac.uk%7C45e97c51cb7d44f6553d08d9a396c025%7C8370cf1416f34c16b83c724071654356%7C0%7C0%7C637720690129293099%7CUnknown%7CTWFpbGZsb3d8eyJWIjoiMC4wLjAwMDAiLCJQIjoiV2luMzliLCJBTil6Ik1haWwiLCJXVCI6Mn0%3D%7C1000&data=i4ljJFLWnRD9Qfrf4B1ZKnhbl3FErEh1PBwxJWb%2Bbql%3D&reserved=0>

We have updated this reference.

I would also consider adding in a reference to the full Liverpool community testing report published on July 7, rather than just a commentary of it as reference 2 is.

The full Liverpool community testing report is cited as reference number 7.

Page 6 line 40-42

See first point about importance of symptom definitions – I would argue “key COVID” symptoms is slightly misleading – many were (officially) ineligible for a PCR test as they did not present with ‘official’ UK gov symptoms, but as headache, fatigue etc are as (or more) common and earlier presenting than eg cough and fever, defining some as “key” and therefore implying others not “key” is questionable. The symptoms most relevant for early identification of cases, and therefore for triaging for testing to exert transmission control are those which are currently not listed on any government or NHS websites.

<https://eur03.safelinks.protection.outlook.com/?url=https%3A%2F%2Fwww.sciencedirect.com%2Fscience%2Farticle%2Fpii%2FS1473309920308379&data=04%7C01%7Cclouise.e.smith%40kcl.ac.uk%7C45e97c51cb7d44f6553d08d9a396c025%7C8370cf1416f34c16b83c724071654356%7C0%7C0%7C637720690129293099%7CUnknown%7CTWFpbGZsb3d8eyJWIjoiMC4wLjAwMDAiLCJQIjoiV2luMzliLCJBTil6Ik1haWwiLCJXVCI6Mn0%3D%7C1000&data=adqeVfZZUm0wnHfO8XuFLTOW2GcN76wXF6BwAXdAfEo%3D&reserved=0>

<https://eur03.safelinks.protection.outlook.com/?url=https%3A%2F%2Fwww.bmj.com%2Fcontent%2F374%2Fbmj.n1625&data=04%7C01%7Cclouise.e.smith%40kcl.ac.uk%7C45e97c51cb7d44f6553d08d9a396c025%7C8370cf1416f34c16b83c724071654356%7C0%7C0%7C637720690129293099%7CUnknown%7CTWFpbGZsb3d8eyJWIjoiMC4wLjAwMDAiLCJQIjoiV2luMzliLCJBTil6Ik1haWwiLCJXVCI6Mn0%3D%7C1000&data=UO%2Bb8Jl%2FcVEqmpFyDObFfvXfil87L%2B3YIUcVF5cNT0%3D&reserved=0>

We agree that the list of COVID-19 symptoms on the NHS and Government websites is not exhaustive. As these are the symptoms that people have been asked to look out for and take a PCR

test and self-isolate in response to, we refer to these symptoms as “COVID-19 symptoms”. To avoid confusion and undue stress on these symptoms, we have now referred to these symptoms as “cardinal COVID-19 symptoms” and listed these symptoms the first time they are mentioned in the manuscript.

Page 7 lines 36-38

Unsure if waves is the best terms here (and in below text), given context of discussion is to mitigate epidemic ‘waves..’ – possibly better to use ‘rounds’?

We have made this amendment.

Page 11 lines 35-42

Is there any data here on how many of the 16.9% took tests as part of asymptomatic testing/screening, rather than because of experiencing symptoms? As this is the key question of this paper here – how many people participating in asymptomatic testing programme – rather than how many people took a lateral flow test in the last week - which cannot be answered without data on symptomatic vs asymptomatic LFT testing.

The aim of this paper was to investigate uptake of lateral flow testing in the population and to identify associated factors. While the Government has launched this as part of an “asymptomatic testing” scheme, we were purely interested in the uptake of testing. As such, we investigated uptake in the sample, regardless of symptom status. We hypothesised that people with symptoms would be more likely to take a test; a result that was confirmed by our data.

Table 2

Was there data to report on difference in reporting testing to Gov/friends&family if tests negative or positive..?

We looked into reporting these data by test result, but numbers for positive or inconclusive test results were very small (negative test n=1063, positive test, n=18, void [inconclusive] n=27, not received results yet n=11, don't know n=4). For this reason, and so not to risk de-anonymising the data, we chose not to report these results.

Page 16 line 9

I think worth reconfirming, and discussing implications of, here also that, of the 17% who took a test in the last week, 65% had taken 2 in the last week (11% of total sample). This suggests even lower than 17% adherence to the policy of biweekly testing (in terms of asymptomatic testing, regular (twice/thrice weekly not one-off)) or pre-event/socialising testing is key to lateral flow tests being effective in reducing transmission, not just a random LFT every now and then..

We have now added a sentence at the start of the discussion, stating that 11% of people report completing at least two LFTs in the last week.

Important to try and distinguish between these: A pre-event, or pre-socialising Friday night test has value (imo), but is different from biweekly testing, and I would suspect engagement and adherence to such testing different, also.

Pre event/pre-socialising testing, and early testing when experiencing symptoms, may be a more realistic policy option which should be considered, and may be a way to increase uptake (and increase benefits realised), rather than asking the public to engage with biweekly testing for months on end even though they have been vaccinated.

We agree that pre-event testing is likely to be one of the major reasons for taking an LFT. However, we did not ask this question in the survey and are therefore unable to report data on this. We have elaborated this point in the limitations.

Page 16 line 33-36

Here again one reason for discrepancy here may be because people more likely to report positive, rather than negative, LFT results. As don't know proportion testing genuinely as part of regular asymptomatic regular testing, or just because they developed (mild or 'official') symptoms, difficult to interpret reasons for differences between data sources here.

Due to the small number of people who reported that their most recent LFT was positive, we were unable to run analyses by test result. People who test positive using an LFT should go on to complete a PCR test. The survey only asks about details of participants' most recent test. For this study, as we were interested in uptake of LFTs, we excluded people's whose most recent test was a PCR test. Other analyses, not reported here, indicate that the percentage of people testing positive was higher in those whose most recent test was a PCR test. However, these results are outside the scope of this manuscript.

Page 17 line 4-6

RE difficulties in adherence and challenges of implementation even when required by employers, I would include at least one of these valuable references from Liverpool pilot:

<https://eur03.safelinks.protection.outlook.com/?url=https%3A%2F%2Facademic.oup.com%2Fageing%2Fadvance-article%2Fdoi%2F10.1093%2Fageing%2Fafab162%2F6322881&data=04%7C01%7Clouise.e.smith%40kcl.ac.uk%7C45e97c51cb7d44f6553d08d9a396c025%7C8370cf1416f34c16b83c724071654356%7C0%7C0%7C637720690129293099%7CUnknown%7CTWFpbGZsb3d8eyJWlloiMC4wLjAwMDAiLCJQIjoiV2luMzliLCJBTiI6IjEhaWwiLCJXVCi6Mn0%3D%7C1000&data=kQnZQCcCdhwm3rBDs%2BsiNZyykAw%2F%2FfP445vMkbu8gA%3D&reserved=0>

<https://eur03.safelinks.protection.outlook.com/?url=https%3A%2F%2Fbmchealthservres.biomedcentral.com%2Farticles%2F10.1186%2Fs12913-021-07191-9&data=04%7C01%7Clouise.e.smith%40kcl.ac.uk%7C45e97c51cb7d44f6553d08d9a396c025%7C8370cf1416f34c16b83c724071654356%7C0%7C0%7C637720690129293099%7CUnknown%7CTWFpbGZsb3d8eyJWlloiMC4wLjAwMDAiLCJQIjoiV2luMzliLCJBTiI6IjEhaWwiLCJXVCi6Mn0%3D%7C1000&data=tu37wcrVNF2nQ189EjwDo0WVG5x2hgjzFnqzQ0I95kk%3D&reserved=0>

Thank you for bringing these to our attention. We have now added a sentence outlining barriers to implementing a testing scheme in the workplace, citing this research.

Page 17 line 10

Disagree with the 'people with symptoms should request a PCR rather than rely on LFT.'
If everyone with relevant symptoms (including symptoms most relevant for identification early enough for transmission control such as headache and fatigue) requested a PCR the system would immediately be swamped.

We have amended this line so that it now reads "UK Government recommendations state that people with cardinal COVID-19 symptoms should request a PCR".

Government policy is that those with symptoms should request a PCR, but this does not mean this is the most optimal way for the system to work (before even get on to cost-effectiveness of continued mass PCR testing when little else being done to control transmission..), and this is a crucial distinction

to make, imo. Authors shouldn't assume those with symptoms 'should' seek a PCR when it is unclear if this is really best public health option.

We have added a new paragraph to the discussion of the manuscript, to further discuss that testing is only effective if used in conjunction with other behavioural interventions (e.g. isolating) and cited research estimating the effect of mass testing schemes alone on the reproduction number for SARS-CoV-2. We have also discussed the effect of widening the symptom definition for eligibility for PCR testing, along with testing burden and case numbers for England since the releasing of restrictions on 19 July 2021.

Concluding remarks

In concluding remarks I think could expand a little more, to include that communications on which test to take and when (and why), and how to act after receipt of a result in a given context, should be far clearer at this stage of the epidemic, and also that testing will never be particularly effective in curbing transmission when there is so little focus on the end to end process (symptom recognition through to action after receipt of the result).

We have added to the concluding remarks that when used in tandem, testing and self-isolation can prevent the spread of COVID-19 by reducing circulation of cases in the community.

Perhaps more could also be said as to whether biweekly testing in a largely vaccinated population is still a proportionate response, especially as uptake is so low to have near negligible effect, and whether this is a distraction from what really matters – should greater focus and resources be given to identify and isolate symptomatic cases (full relevant symptom range) early, and also increase engagement with pre-event/pre-socialising testing. If system optimised towards these individuals where impact on epidemic greatest, rather than incredibly poorly implemented mass screening of low risk populations, benefits likely to outweigh biweekly testing with 11% uptake..

We have added a sentence to the discussion indicating that estimates of the effectiveness of mass asymptomatic testing were conducted before widespread rollout of vaccination. We have also added to the concluding remarks that consideration should be given to how best to optimise the testing system (PCR and LFT), taking into account the full range of symptoms displayed early in SARS-CoV-2 infection, transmission when asymptomatic, financial cost of testing, and burden on the testing system.

VERSION 2 – REVIEW

REVIEWER	Green, Mark University of Liverpool, Geography & Planning
REVIEW RETURNED	06-Dec-2021

GENERAL COMMENTS	Thank you to the authors for their hard work in making the revisions. I am happy to accept the paper as is - the paper is a nice output and I look forward to seeing it published. Well done!
---

REVIEWER	Nguyen, Le Khanh Ngan Strathclyde Business School Department of Management Science
REVIEW RETURNED	03-Jan-2022

GENERAL COMMENTS	The authors addressed all my concerns well, and I recommend the manuscript for publication.
---